# The conformational plasticity of structurally unrelated lipid transport proteins correlates with their mode of action

**Sriraksha Srinivasan**[1][☉], **Andrea Di Luca**[1][☉], **Daniel Álvarez**[1,2], **Arun T. John Peter**[1], **Charlotte Gehin**[3], **Museer A. Lone**[4], **Thorsten Hornemann**[4], **Giovanni D'Angelo**[3], **Stefano Vanni**[1,5]*

**1** Department of Biology, University of Fribourg, Fribourg, Switzerland, **2** Departamento de Química Física y Analítica, Universidad de Oviedo, Oviedo, Spain, **3** Institute of Bioengineering (IBI) and Global Heath Institute (GHI), École Polytechnique Fédérale de Lausanne (EPFL), Lausanne, Switzerland, **4** Institute of Clinical Chemistry, University Hospital Zurich, University of Zurich, Zurich, Switzerland, **5** National Center of Competence in Research Bio-inspired Materials, University of Fribourg, Fribourg, Switzerland

☉ These authors contributed equally to this work.
* stefano.vanni@unifr.ch

**Data Availability Statement:** Input files for atomistic and coarse-grained MD simulations, structures of the LTD with the membrane binding interface, and representative apo-like and holo-like

## Abstract

Lipid transfer proteins (LTPs) are key players in cellular homeostasis and regulation, as they coordinate the exchange of lipids between different cellular organelles. Despite their importance, our mechanistic understanding of how LTPs function at the molecular level is still in its infancy, mostly due to the large number of existing LTPs and to the low degree of conservation at the sequence and structural level. In this work, we use molecular simulations to characterize a representative dataset of lipid transport domains (LTDs) of 12 LTPs that belong to 8 distinct families. We find that despite no sequence homology nor structural conservation, the conformational landscape of LTDs displays common features, characterized by the presence of at least 2 main conformations whose populations are modulated by the presence of the bound lipid. These conformational properties correlate with their mechanistic mode of action, allowing for the interpretation and design of experimental strategies to further dissect their mechanism. Our findings indicate the existence of a conserved, fold-independent mechanism of lipid transfer across LTPs of various families and offer a general framework for understanding their functional mechanism.

## Introduction

Lipids are one of the key building blocks of eukaryotic cells, as they allow for the spatial and temporal organization of chemical reactions in different cellular compartments called organelles. Eukaryotic cells contain thousands of different lipid types, and each membrane-bound organelle possesses a characteristic lipid composition necessary for its proper functioning [1]. This compositional identity is crucial not only towards their functions but also to shape and regulate intracellular signaling and trafficking processes between them [2].

conformations of each LTD arising from clustering of the atomistic simulations, as well as the data presented in the Figures, protocols and scripts can be found at: https://doi.org/10.5281/zenodo.12728271.

**Funding:** S.V. acknowledges support by the SNSF (PP00P3_194807) and by the European Research Council under the European Union's Horizon 2020 research and innovation program (grant agreement no. 803952). G.D'A. acknowledges support by the Swiss Cancer League, KFS-4999-02-2020; by the EPFL institutional fund; and by SNSF (310030_184926). This work was supported by grants from the Swiss National Supercomputing Centre under projects ID s1030 and s1132. S.V. and A.J.P. acknowledge support from the Novartis Forschungsstiftung via a FreeNovation grant. D.A. acknowledges support from the Margarita Salas program 2021–2023 funded by Ministerio de Universidades (MU-21-UP2021-030-53773022). M.A.L. acknowledges support from the Foundation Suisse de Recherche sur le Maladies Musculaires (FSRMM). TH acknowledges support from Schweizerischer Nationalfonds zur Förderung der Wissenschaftlichen Forschung (310030_215134) and from European Joint Program on Rare Diseases (32ER30_187505). MAL acknowledges support from EMPIRIS foundation, Zürich. The funders had no role in study design, data collection and analysis, decision to publish, or preparation of the manuscript.

**Competing interests:** The authors have declared that no competing interests exist.

**Abbreviations:** CG, coarse grain; DOG, dioleoylglycerol; DOPC, dioleoyl-phosphatidylcholine; ER, endoplasmic reticulum; KL, Kullback–Leibler; LTD, lipid transport domain; LTP, lipid transfer protein; MD, molecular dynamics; PCA, principal component analysis; PH, pleckstrin homology; PME, Particle Mesh Ewald; RMSD, root mean square deviation; RMSF, root mean square fluctuation; VDW, van der Waals; wt, wild type.

Since lipid synthesis is not ubiquitous, but rather mostly localized to the endoplasmic reticulum (ER) [3], lipids must be rapidly transported between organelles to maintain lipid homeostasis and organellar identity. This is achieved via 2 main routes, the vesicular and nonvesicular pathways. In the vesicular pathway, cargo vesicles, originating from lipid remodeling processes mediated by coat proteins, travel from a donor organelle to an acceptor one, where the vesicle undergoes fusion [4]. This pathway is not only crucial for cellular exocytosis and endocytosis but also intracellularly along the secretory pathway [4]. Alternatively, in the nonvesicular pathway, trafficking of lipids between organelles is performed by lipid transfer proteins (LTPs), which solubilize lipids and facilitate their transport between 2 membranes. Nonvesicular lipid transport promotes a more rapid modulation of the lipid composition of organelles compared to vesicular trafficking and is crucial during stress conditions when vesicular trafficking is compromised [5].

Despite growing interest in the nonvesicular lipid transport pathway, our mechanistic understanding of how LTPs perform their function is still largely incomplete. The only unifying feature of LTPs is a polar exterior and the presence of a lipid transfer domain (LTD) containing a hydrophobic cavity that encloses the lipid. This architecture, by shielding the hydrophobic lipid molecule from the aqueous environment of the cytoplasm, reduces the energetic cost of transferring a lipid between 2 membranes. Two main models of lipid transport by LTPs have been put forward: the shuttle model and the tunnel model. In the shuttle model, a small (<50 to 100 kDa) LTD travels between the donor and the acceptor organelle, cyclically taking up and releasing their substrate lipid [5]. In the tunnel model, a large (>100 kDa) LTD physically connects the 2 organelles, establishing a continuous hydrophobic pathway in which lipids can simply diffuse between the 2 membranes [5]. In both cases, however, a fine regulation of multiple mechanistic steps must be accurately tuned to achieve lipid transport with the correct directionality and rate, and how such complex coordination is achieved remains largely unclear [5].

This picture is further complicated by the sheer number of existing LTPs. So far, hundreds of different LTPs, belonging to several distinct protein families [6,7], have been identified. This diversity possibly originates from the huge chemical variability of the lipid substrates and organellar membranes they bind to. While this has likely allowed fine-tuning the mechanism and specificity of lipid transport by LTPs, it has so far prevented the establishment of a common framework to understand and interpret the molecular steps underlying this process. To this extent, only a few studies have attempted to investigate in a high-throughput fashion the functional properties of LTPs, such as their membrane or lipid binding [8,9]. Rather, the investigation of individual LTPs using cellular biology or reconstitution approaches remains to date the most frequent strategy.

A direct consequence of this case-by-case modus operandi is that a plethora of concurring models have been put forward to explain the mechanism and specificity of lipid transport. Yet, these models are of limited transferability across different LTPs since they largely rely on specific observations either on protein structure (such as the presence of a lid [6,10], of electrostatic surface patches [11,12], or on the specificity of the lipid-binding cavity [13]) or on experimentally determined transport properties (such as counter-exchange between different lipid species and lipid-dependent transport rates) [14,15].

While these studies were limited to individual LTPs, when viewed altogether, they suggest features that could be shared among many of them. Specifically, a potential role for LTP conformational plasticity, i.e., their ability to adopt multiple conformations, has been proposed for various LTPs belonging to distinct protein families, including Osh/ORP, Ups/PRELI, PITP, and START family members [11,13,16–29]. These works have proposed that conformational changes between apo-like and holo-like structures might regulate lipid uptake and release by

individual LTPs, but whether this is a general mechanism that extends to most LTPs remains unexplored.

Here, we employ a computational-based alternative approach to characterize the conformational plasticity of multiple LTDs belonging to different families. We find that despite significant differences in 3D structure, small LTDs (<50 to 100 kDa) can adopt distinct conformations based on the presence or absence of a bound lipid. The ability to transition between these conformations has potential implications for their functional mechanism, including membrane binding, and suggests a conserved mechanism of lipid transfer across LTPs of diverse families.

## Results

### The membrane binding interface of lipid transport domains does not display any conserved structural signature

To identify common, family-independent properties of LTPs, we opted for an in silico analysis based on molecular dynamics (MD) simulations. To this end, we simulated and analyzed 12 LTDs belonging to 8 different families, ensuring a sampling of proteins with diverse secondary structures and partner lipids (Fig 1A). Specifically, we selected all LTPs for which a crystallographic structure in complex with a lipid was available at the start of our project, plus 2 additional proteins (GM2A and LCN1) to increase the structural diversity of our dataset (Fig 1A).

Since LTDs uptake lipids from membranes of cellular organelles, we first investigated how different LTDs bind to model lipid bilayers, using a computational protocol we recently developed [30]. In short, to determine the membrane-binding interfaces of LTDs, we performed coarse grain (CG) simulations of the proteins in combination with pure dioleoyl-phosphatidyl-choline (DOPC) lipid bilayers, starting with the protein at a minimum distance of 3 nm away from the bilayer (Fig 1B). Over the course of the MD simulation, the proteins exhibit transient interactions, i.e., multiple binding and unbinding events, as shown by the minimum distance curves in Fig A in S1 Text. The membrane-interacting residues of each protein are then determined by computing the frequency of the interaction of each residue with the bilayer (see Methods). Notably, the proteins in our dataset display distinct binding affinities, with some proteins showing very transient binding while others remain membrane-bound for most of the simulation trajectory (Fig A in S1 Text). This behavior could be, in part, attributed to the wide diversity of organellar membranes to which the LTDs in our dataset bind to in vivo, and to the comparative simplicity of our in silico model lipid bilayers.

Fig 1C shows the structure of each protein in our dataset, colored by the frequency of interaction with the lipid bilayer. The corresponding residue-wise analysis of the frequency of interactions is reported in Fig B in S1 Text. Despite intrinsic limitations of our CG simulations for what pertains the description of protein conformational changes, the good agreement between the membrane interface determined from the simulations and the experimentally-proposed one available for CPTP [31], FABP [32], Osh4 [21], Osh6 [11] GRAMD1A [27], GRAMD1C [27], PITPA [33], and TTPA [34] (Fig B in S1 Text) suggests that our in silico methodology is mostly able to reproduce the correct membrane-binding interface of LTDs, similar to what was shown for different families of membrane-binding peripheral proteins [30].

Using the obtained residue binding propensity, we next investigated whether LTDs belonging to different families possess common structural membrane binding signatures. First, we investigated whether the membrane binding interface displays specific sequence properties (Fig 1D and Fig C in S1 Text). Concomitant analysis of all LTDs (Fig 1D) indicates that the membrane binding interface of LTDs is enriched in the positively charged amino acid Lysine, as this amino acid is less membrane-disruptive than Arginine [35], and aromatic/hydrophobic

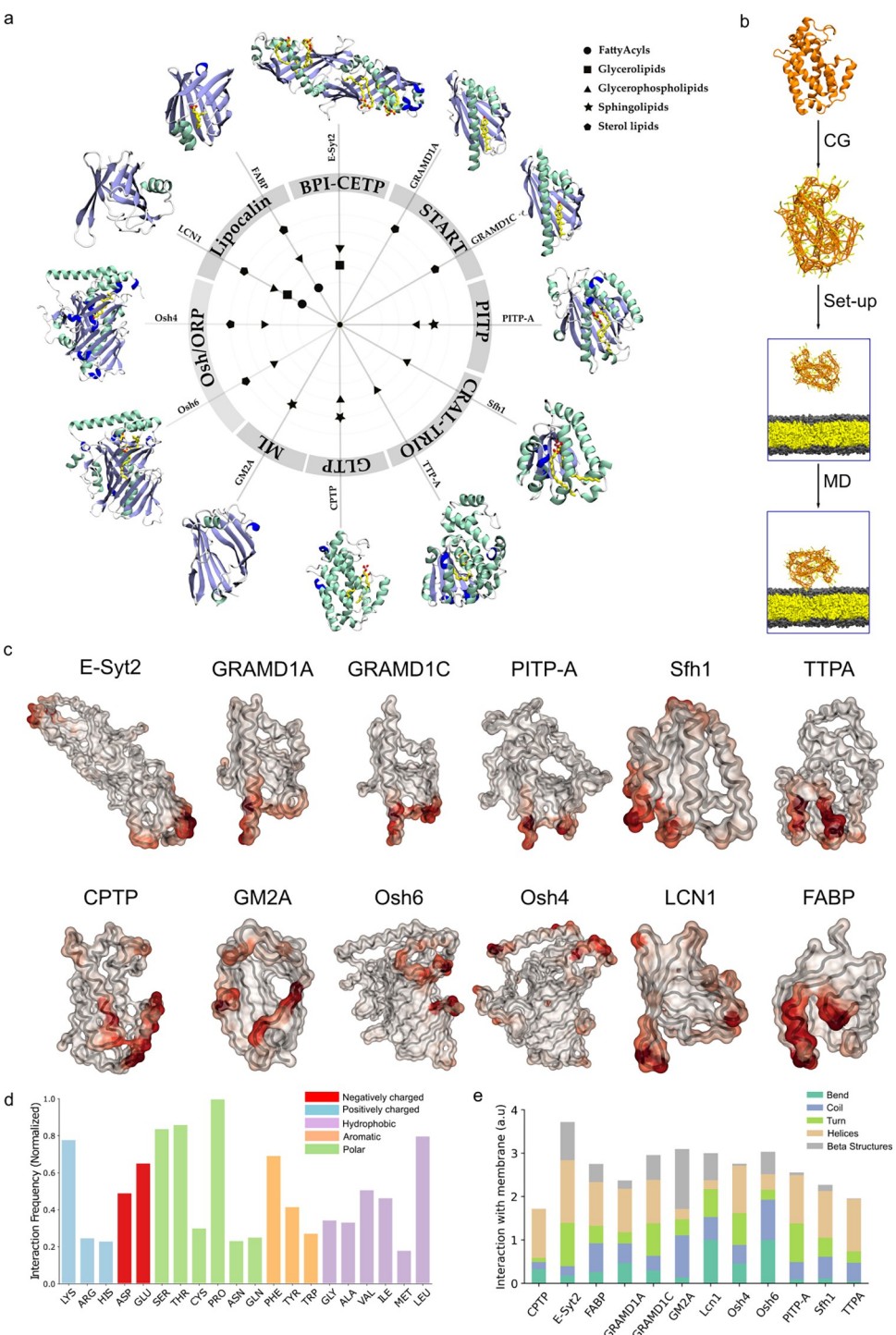

**Fig 1. Membrane-binding regions of LTPs do not display shared structural features.** (**a**) Members of 8 different LTP families used in this study. Symbols inside the circle indicate their lipid specificity [7]. The secondary structures of the LTDs are shown along the periphery. (**b**) Schematic of the protocol followed to perform MD simulations of LTD-bilayer systems and analyze interacting regions (**c**) LTD structures colored by their frequency of interaction with lipid bilayers (**d**) Interaction frequency of each amino acid with the membrane, summed over all LTDs in the dataset (**e**) Interaction frequency of each secondary structure type. LTD, lipid transport domain; LTP, lipid transfer protein; MD, molecular dynamics.

ones (Phe, Leu, Val, Ile). This confirms previous observations, as (i) binding of negatively charged lipids via positively charged residues and (ii) hydrophobic insertions are 2 of the main mechanisms involved in membrane binding by peripheral proteins [35–40]. Most notably, also polar residues such as Ser, Thr, and Pro seem to be involved in membrane binding by LTDs. However, analysis on a protein-by-protein scale reveals a lack of any general trend, as each protein family displays a characteristic amino acid composition in its membrane binding interface (Fig C in S1 Text), possibly as a consequence of the diversity of the organellar membrane they bind to (Table 1). Similarly, analysis of the secondary structure of the membrane binding interface does not display any preference for secondary structure elements (Fig 1E).

## The membrane binding interface of lipid transport domains is characterized by large collective motions

We next investigated whether dynamical properties, rather than structural ones, could explain, or correlate with, membrane binding in LTDs. To do so, we performed extensive atomistic MD simulations of the proteins in solution, and we characterized their dynamical signature.

To this end, we first characterized LTDs' local fluctuations by computing their root mean square fluctuations (RMSFs), and we combined this analysis with our membrane-binding assay (Fig 2A). Notably, we observed that almost all LTD membrane binding regions (red bars, Fig 2B) correspond to maxima of RMSF (blue curves, Fig 2B). To further quantify the relationship between residues' RMSF and membrane interaction frequency, we computed the RMSF for 2 distinct residues' populations: those that do interact with the membrane versus those that do not (Fig 2C). Comparison of the 2 groups suggest that this difference is statistically significant for all proteins, with the sole exception of Sfh1. Overall, these data indicate that highly dynamical protein regions are often located at or near the membrane-binding interface regardless of the overall protein fold and structure. This observation agrees with general analyses performed on other peripheral proteins and their membrane binding motifs, which identified protruding loops as a main driver for protein–membrane interactions [35].

Since RMSF is mostly indicative of thermal fluctuations and local protein disorder, we next sought to investigate whether membrane binding is also related to large-scale protein motions. To this end, we analyzed the MD simulations using principal component analysis (PCA) (Fig 3), as this technique can be used to capture dominant, large-scale variations in protein conformations sampled during the MD simulations [41,42]. We specifically focused on PC1 as it explains most of the variance in the dynamics (38% on average for all the proteins in our dataset; see Table A in S1 Text). Using this approach, we could identify the protein regions undergoing the largest collective motion (Fig D in S1 Text).

We combined this analysis with our membrane-binding assay (Fig 3) to investigate whether membrane-binding regions are also involved in the proteins' principal motions. Notably, we observed that almost all LTD membrane binding regions (red bars, Fig 3A) correspond to maxima of the contributions of single-residue dynamics to PC1 (blue curves, Fig 3A). As for RMSF (Fig 2C); analysis of the relationship between residues' PC1 contribution and membrane interaction frequency for membrane interacting (Fig 3B, orange bars) versus noninteracting (Fig 3B, red bars) residues indicates that membrane binding residues have significantly more contribution towards PC1 for almost all proteins (Fig 3B). This indicates that protein regions located at or near the membrane-binding interface are involved in large collective motions regardless of the overall protein fold and structure.

In our dataset, exceptions to this rule seem to be Sfh1, LCN1, and Osh6 (Fig 3B). In the case of Osh6, however, the experimentally determined membrane-binding region at the N-terminus [11] is only marginally binding to the lipid bilayer in silico, and it also appears to have

**Table 1. LTDs used for MD simulations.**

| Protein | PDB ID | Residues of LTD used (Uniprot numbering) | Bound-lipid in atomistic simulations of holo-form | Localization to intracellular organelles |
|---|---|---|---|---|
| CPTP | 4KBS (apo), 4K8N (holo) | 8–214 <br> 8–214 | Ceramide-1-phosphate | ONM, Endosome, PM, Golgi |
| E-Syt2 | 4P42 | 191–370[a] (chain A and B) | Dioleoylglycerol (2 molecules per chain) | PM, ER |
| FABP | 5CE4 | 2–132 | Oleic acid | - |
| GRAMD1A | 6GQF | 364–536 | Cholesterol | ER, PM, Autophagosome, Cytoplasmic vesicles |
| GRAMD1C | 6GN5 | 323–499 | Cholesterol | ER, PM |
| Osh4[b] | 1ZHZ | 1–434 | Ergosterol | ER, Golgi, Mitochondria |
| Osh6 | 4B2Z | 36–434 | DOPS | ER, PM |
| PITPA | 1T27 | 2–270 | DOPC | - |
| Sfh1 | 3B74 | 101–274[c] | DOPE | Golgi |
| TTPA | 1OIP | 88–253[c] | α-tocopherol | - |
| GM2A | 1PU5 | 33–193 | - | Lysosome |
| LCN1 | 1XKI | 30–168 | - | - |
| STARD11 | 2E3R | 364–624 | - | ER, Golgi |
| CPTP-V158N | 4KBS (apo), 4K8N (holo) | 8–214 <br> 8–214 | Ceramide-1-phosphate | - |
| GRAMDIA-5P | 6GQF | 364–536 | Cholesterol | - |
| PITPA-P78L | 1T27 | 2–270 | DOPC | - |
| STARD11-5P | 2E3R | 364–624 | - | - |

DOPC, dioleoyl-phosphatidylcholine; DOPE, dioleoyl-phosphatidylethanolamine; DOPS, dioleoyl-phosphatidylserine; ER, endoplasmic reticulum; ONM, outer nuclear membrane; PM, plasma membrane.

[a]Only one of the 2 chains was considered for analysis of the atomistic simulations.

[b]The PDB structure of Osh4 in complex with cholesterol (1ZHY) was used for coarse-grained simulations with the bilayer.

[c]Additional residues (Sfh1: 4–310 and TTPA: 25–275) were retained in the simulations in order to maintain the secondary structure of the protein but were not considered for the analysis since they do not constitute the CRAL-TRIO lipid transfer domain.

The loop missing from the crystal structures deposited in the RCSB PDB structure 5GYD was rebuilt by adding 5 residues to each truncated side of the structure. The loop has been shown to be dispensable for function [81].

limited contribution to PC1. This is likely because our simulations are unable to sample the large conformational changes that the N-terminal lid of Osh6 has been proposed to undergo from its lipid-bound to its apo state, indicating that insufficient sampling could be the reason for this apparent discrepancy. For LCN1, on the other hand, its low membrane affinity in CG simulations (Fig A in S1 Text) results in a large number of membrane-binding residues, effectively making this analysis very noisy.

## Lipid transport domains in solution display a conformational equilibrium that is modulated by the presence of bound lipids

Since our data indicate a relationship between LTDs' local and collective motions and membrane binding (Figs 2 and 3), we next opted to further characterize the conformational landscape of LTDs. To do so, we computed the population distribution of the projections of the

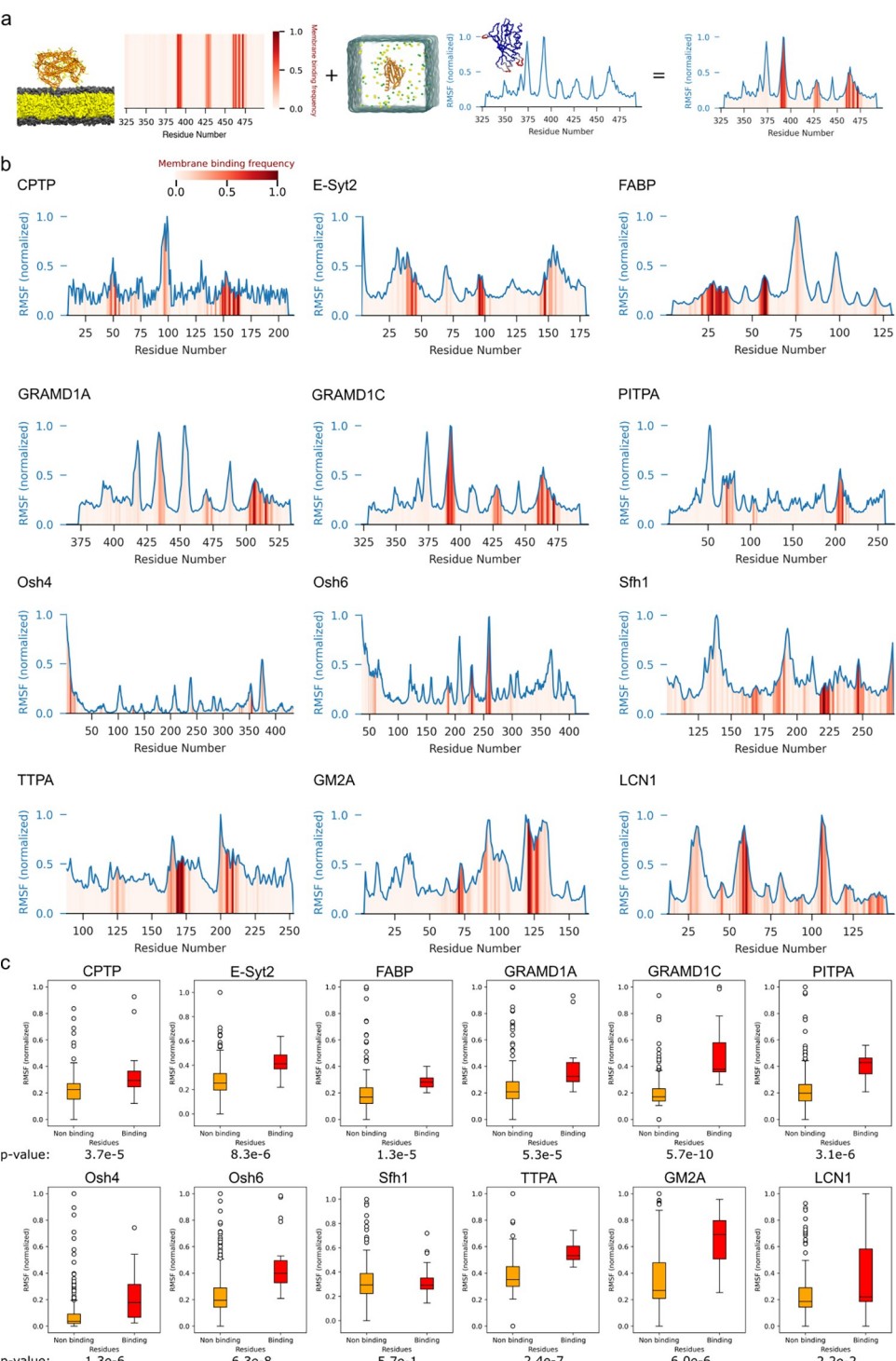

**Fig 2. Membrane-interacting regions of LTDs correlate with their dynamical regions. (a)** The frequency of interaction of each residue with the lipid bilayer is represented by the heatmap in red, while each residue RMSF is indicated by the line plot in blue. Both measures have been normalized to a value between 0 and 1 for each protein, respectively. **(b)** Box plots showing the distribution of RMSF values for membrane binding and nonbinding residues. *P* values show the difference between both distributions, computed with a Mann–Whitney U test. The data underlying the graphs shown in the figures can be found in https://doi.org/10.5281/zenodo.12728271.

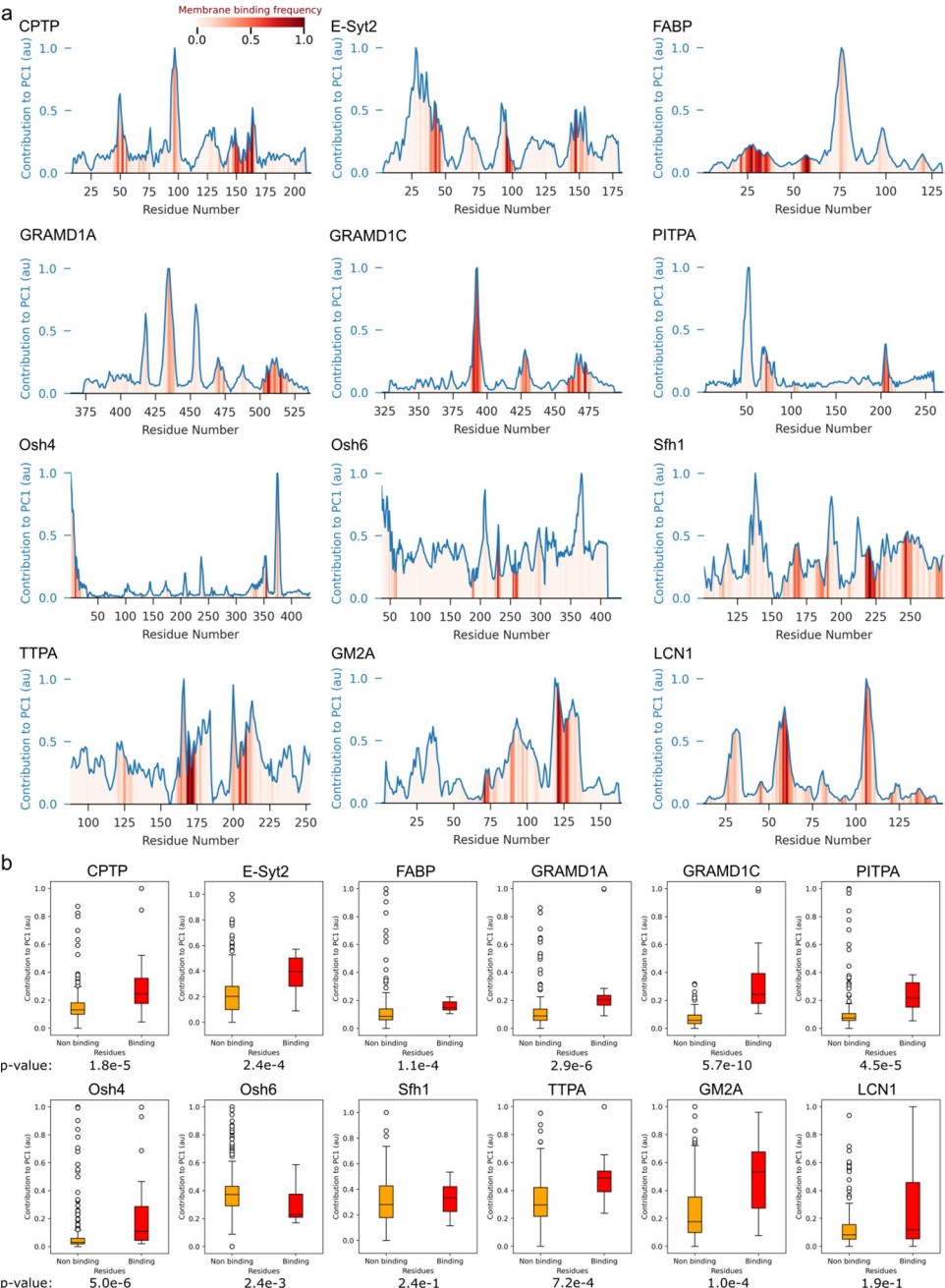

**Fig 3. Membrane-interacting regions of LTDs correlate with their dynamical regions. (a)** Schematic of the combination of membrane-binding interface of the protein from CG-MD simulations and the dynamics of the protein in water from atomistic MD simulations **(b)** The frequency of interaction of each residue with the lipid bilayer is represented by the heatmap in red, while the contribution of each residue to the first principal component of protein dynamics (PC1) is indicated by the line plot in blue. Both measures have been normalized to a value between 0 and 1 for each protein, respectively. **(c)** Box plots showing the distribution of the contribution to PC1 values for membrane binding and nonbinding residues. *P* values show the difference between both distributions, computed with a Mann–Whitney U test. The data underlying the graphs shown in the figures can be found in https://doi.org/10.5281/zenodo.12728271. CG, coarse grain; LTD, lipid transport domain; MD, molecular dynamics.

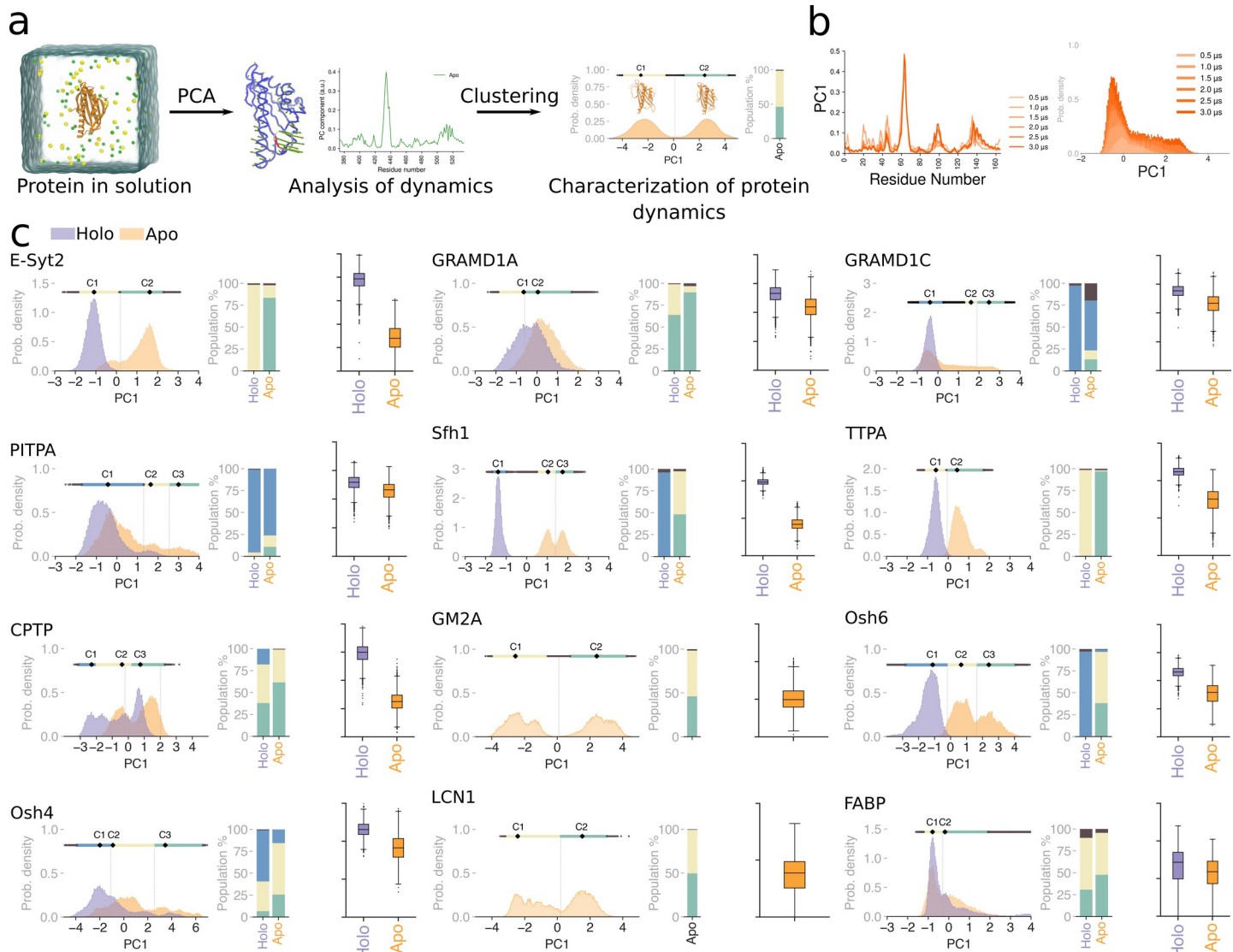

**Fig 4. Bound lipids modulate the conformational landscape of LTDs. (a)** Protocol to characterize protein dynamics from atomistic simulations of the protein in solution. The first principal component PC1 was determined, and the resulting conformations were clustered using a density-based automatic procedure. **(b)** Simulations of 6 replicas for 500 ns each were performed to converge the distribution of PC1. **(c)** The population distributions of PC1 from apo (orange) and holo (purple) simulations of the protein are shown. The clusters of the distributions are indicated by the line above the histogram, with the black dot (labelled C1, C2, and C3) representing the cluster centers. Bar plots in the center indicate the relative apo and holo populations of each cluster. Box plots on the right indicate cavity volumes for apo (orange) and holo (purple) forms of the protein. Holo-forms of the protein could not be simulated for GM2A and LCN1 due to the lack of lipid-bound crystal structures. The data underlying the graphs shown in the figures can be found in https://doi.org/10.5281/zenodo.12728271.

first principal component (PC1) for all LTDs in our dataset, and we clustered the resulting conformations using a density-based automatic procedure [43] (Fig 4A). Simulations of 6 replicas for 500 ns each were performed to converge the distribution of PC1. Our results indicate that in the apo form, the proteins sample a diverse conformational landscape as shown by the multimodal distribution of the populations of PC1 (Fig 4C, orange histograms). Notably, and despite significant differences in the conformational landscape of the various LTDs, density-based automated clustering is able to distinguish at least 2 distinct clusters for each protein (Fig 4C, bar plots).

The alternation between different conformations is a hallmark of enzyme dynamics, where it generally correlates with protein activity [44]. To push further this parallelism, we next investigated the effect of the bound lipid on an LTD's conformational landscape, since in classical enzymology the presence of a bound substrate is generally known to restrict enzyme dynamics and stabilize the protein in a specific conformation [45].

To determine if the presence of the bound lipid would alter the conformational preference of the LTDs in our dataset, for all cases in which a bound lipid was co-crystallized together with the protein (10 out of 12 proteins in our dataset), atomistic simulations in the holo-form were performed and analyzed following the same protocol used for the apo-forms. Notably, the comparison of the projections along the PC1 from the holo-form with that of the apo-form shows that, for all proteins, the presence of the bound lipid shifts the population distributions along the PC (Fig 4C, purple histograms). The residue-wise contribution to PC1 for simulations in the apo form, holo form, and when taken together is shown in Fig D in S1 Text. In detail, the presence of a bound lipid appears to stabilize the protein in one specific conformation (Fig 4C, bar plots). In most cases, this conformation is also sampled by the protein in its apo form, but always with a lower frequency than the corresponding holo simulations, indicating that the presence of a bound lipid indeed "locks" the LTD in a specific 3D structure, akin to substrate binding in enzyme dynamics.

To further characterize this conformational landscape, we next evaluated the structural properties of representative conformers belonging to the 2 main clusters emerging from the PC analysis. To do so, we first computed the root mean square deviation (RMSD) of structures corresponding to the 2 extremes of the clusters along the PC1 spectrum (Fig E in S1 Text). For the LTDs in our dataset, this value varies between 2.3 and 7.1 Å, indicating that the range of structural difference between clusters is highly variable.

Next, we computed the volume of the hydrophobic cavity of the structures sampled in our MD simulations for both apo and holo trajectories. For almost all LTDs investigated here, we could quantify significant differences in cavity volume for the 2 conditions (apo versus holo) (Fig 4C and Fig F in S1 Text). In almost all cases, when no lipid is found inside the protein, its hydrophobic cavity shrinks, thus reducing its size (Fig 4C and Fig F in S1 Text). Consistent with the behavior of the entire protein (Fig 4C), the cavity in the holo state generally adopts a more compact distribution of conformations (Fig F in S1 Text, purple), while, in the absence of bound lipids, the cavity exhibits larger variance in its volume (Fig F in S1 Text, orange).

## Lipid transfer rates are affected by LTD conformational plasticity in apo and holo states

Our data suggest that all LTDs exist in an equilibrium between 2 or more conformations and that the presence of bound lipids alters their equilibrium population, akin to a mechanism of "conformational selection" or "induced fit." For all LTDs, these conformational changes are mostly localized in protein regions that interact with the lipid bilayer. Since the main proposed activity of LTDs is to transport lipids, we next sought to investigate whether this observed "enzyme-like" behavior has functional consequences.

We first sought to investigate whether the observed conformational plasticity of LTDs can provide clues into experimentally determined lipid transport rates. To this extent, the 2 members of the ASTER family in our dataset provide a fertile ground for this analysis, since the ASTER domains of GRAMD1A and GRAMD1C have very similar structures (RMSD X-ray = 0.76 Å) yet very different transport rates [46], with GRAMD1A transporting approximately 8.4 and GRAMD1C between approximately 0.5 and 1.5 DHE molecules/min/molecule LTP in identical experimental conditions in vitro (Fig 5A) [46]. Notably, despite their

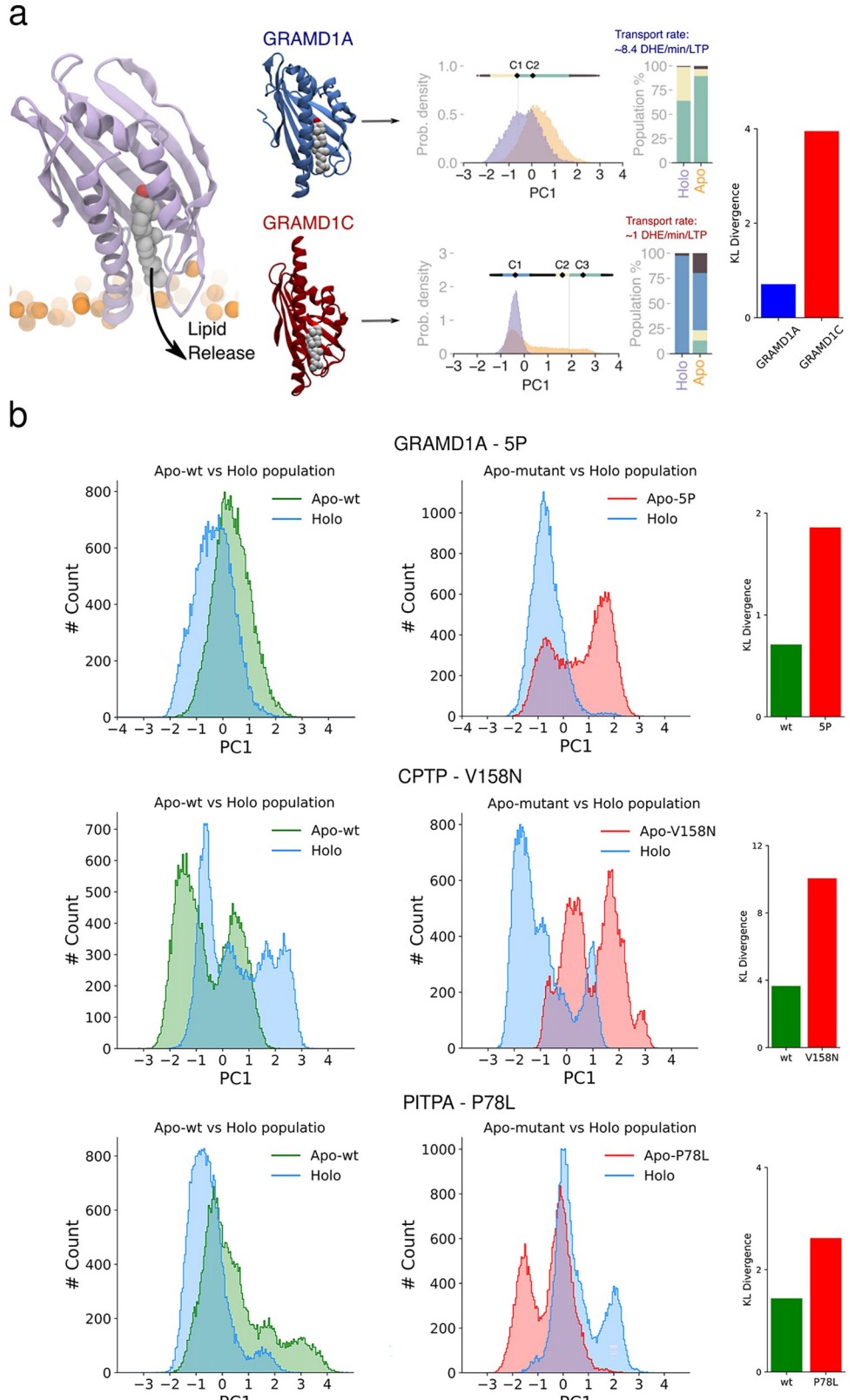

**Fig 5. Overlap between apo and holo populations of LTPs can influence their transport rates. (a)** The difference in transport rates between GRAMD1A and GRAMD1C can be related to their distinct conformational equilibria. Left:

3D structure of GRAMD1A and GRAMD1C. Center: population distributions of PC1 from apo (orange) and holo (purple) forms of GRAMD1A and GRAMD1C, with corresponding DHE transport rates. Right: KL divergence between apo and holo for GRAMD1A (blue) and GRAMD1C (red). **(b)** Mutations of GRAMD1A (5P), CPTP (V158N), and PITPA (P78L) that decrease their transport rates in vitro also reduce the similarities between apo and holo distributions. Left: population distributions of PC1 from wt-apo (green) and holo (blue). Center: population distributions of PC1 from mutant-apo (red) and holo (blue). Right: KL divergence between wt-apo and holo (green) and mutant-apo and holo (red). Concatenating the trajectories between Holo and Apo-wt vs. Holo and Apo-mutant results in slight differences between the holo population distribution in the 2 graphs. The data underlying the graphs shown in the figures can be found in https://doi.org/10.5281/zenodo.12728271. KL, Kullback–Leibler; LTP, lipid transfer protein; wt, wild type.

structural similarities, the 2 proteins display markedly different conformational populations in our dynamical analysis, with GRAMD1A exhibiting a significantly more overlapped population between apo and holo states compared to GRAMD1C (Fig 5A). To quantify the relationship between apo and holo PC1 distributions, we computed the average Kullback–Leibler (KL) divergence between the 2 states (the closer to 0 the KL value, the more similar the 2 distributions), and we found that the KL divergence is of 0.71 for GRAMD1A and 3.96 for GRAMD1C, further highlighting the much more similar profile of the apo-holo distributions for GRAMD1A. Overall, this observation correlates well with the ability of GRAMD1A, but not of GRAMD1C, to interchange between different conformations, suggesting that overlap between apo and holo states could promote lipid transport activity.

To further investigate the relation between LTP's conformational plasticity and lipid transport, we next identified 3 LTD mutants of proteins in our database that have been characterized to decrease lipid transport rates experimentally (GRAMD1A 5P [47], CPTP V158N[12] and PITP P78L [48]). For these proteins, we performed MD simulations of the mutant apo and holo forms to investigate whether introducing mutations alters the conformational distribution of these states and to what extent it affects the overlap between apo and holo populations. To do so, we computed the KL divergence between the apo form of these mutants and their respective wild-type holo form, which we take as "ground truth" for the transport activity (Fig 5B), as well as the KL divergence between the apo and holo mutant forms (Fig G in S1 Text). These analyses confirm that for all mutants, which are defective in lipid transfer, KL divergence increases with respect to the wild type (Fig 5B), with the sole exception of the GRAMD1A 5P when holo and apo mutants are compared (Fig G in S1 Text). However, it is likely that this apparently negative result originates from the large effect of the 5P mutation on protein conformational equilibrium, as both apo-mutant and holo-mutant conformational distributions deviate significantly from those of the wild type (Fig 5B and Fig G in S1 Text). This results in enhanced destabilization of the lipid-bound state, hence yielding a conformation that is not conducive to lipid transport. Furthermore, there is no indication that holo-mutant conformations represent bona fide thermodynamic minima, as the initial conformations for these mutants can only be generated starting from the wild-type structures (see Methods), and considering that these mutants are heavily defective in lipid transport might not bind and transport lipids in native conditions.

Taken together, these results further suggest that the ability to interconvert between apo and holo conformations might impact lipid transfer rates.

## The conformational equilibrium of lipid transport domains can modulate their lipid transport activity in cells

Next, we investigated whether mutations able to alter the conformational dynamics of LTDs, as suggested by our MD simulations, would also impair lipid transport in cellulo. To do so, we selected a protein that was not in our original dataset, but for which lipid transport activity has

been extensively characterized in cellular assays: STARD11 (also known as CERT), a member of the START family [49].

STARD11 is an LTP that promotes ceramide transport from the ER to the *trans*-Golgi at ER-Golgi membrane contact sites. It is composed of an N-terminal pleckstrin homology (PH) domain mediating its anchoring to the *trans*-Golgi [50,51], 2 phenylalanines in an acidic tract (FFAT) motif responsible for binding to the ER membranes [50], and a C-terminal and a (StAR)-related lipid transfer (START) domain that extracts ceramide from the ER membrane and delivers it to the *trans* Golgi. Once transported by STARD11 to the *trans*-Golgi, ceramide is readily converted into sphingomyelin [52].

To alter the conformational dynamics of the START domain of STARD11, we mutated the Ω1 loop (Fig 6A), which is involved in the principal motion of the protein for both STARD11 (Fig H in S1 Text) as well as for the other START-containing LTDs in our dataset, GRAMD1A and GRAMD1C (Fig 3). Analogously to what is proposed for ASTER proteins [47], we opted to mutate residues 497 to 501 (RVWPA, which are residues 471 to 475 in the crystal structure) into a stretch of prolines (for short, STARD11-5P) (Fig 6A). As expected, this mutation results in a significant shift of the population distribution of STARD11 along the first principal component (Fig 6B), resulting in a KL divergence shift from a value of 0.67 for the wild type to 1.78 for the STARD11-5P mutant (Fig 6B).

Next, to test STARD11-5P bioactivity, we first expressed GFP-tagged STARD11-5P and compared its intracellular localization with that of STARD11 wild type (wt). Intriguingly, we found that STARD11-5P is significantly more membrane associated than STARD11 wt (Fig 6C and 6D). STARD11 membrane association is controlled by its phosphorylation status with hyper- and hypo-phosphorylated forms of STARD11 being more cytosolic and membrane associated, respectively [53,54]. Accordingly, we found that STARD11-5P is hypo-phosphorylated relative to STARD11 wt (Fig 6E).

Modulation of STARD11 phosphorylation/localization is part of a homeostatic circuit whereby excessive STARD11 activity and sphingomyelin production trigger a signaling reaction leading to its phosphorylation and inactivation [53,55,56]. STARD11-5P appears to be unable to induce this response possibly due to its inability to sustain ceramide transfer and conversion to sphingomyelin. To test this hypothesis, we evaluated the sphingomyelin levels in STARD11-KO HeLa cells [52] expressing either STARD11 wt or STARD11-5P. We found that while expression of STARD11 wt rescues the defect in sphingomyelin production in STARD11-KO HeLa cells, STARD11-5P fails to do so (Fig 6F). Thus, impairing the dynamics of the START domain of STARD11 inhibits its bioactivity.

Taken together, these observations suggest that the conformational alterations observed in our MD simulations upon mutations in STARD11 are physiologically relevant, and further support our model that the conformational plasticity of LTPs is important for their activity and cellular function.

## Discussion

Based on experimental observations on individual LTPs [11,27,31,33,34,57,58], several concurring models have been put forth in recent years to elucidate their mechanism of action [6,59,60]. Even though these models have highlighted active-site residues and membrane binding regions or protein sequences that act as potential gates for lipid uptake/release, a unifying model to explain LTP function in a global context remains lacking. Building on this limitation, in this work, we used MD simulation to investigate several LTPs concurrently to decipher the common thread in their mechanism of action. First, using our previously established protocol to study peripheral protein–membrane interactions, we determined the membrane binding

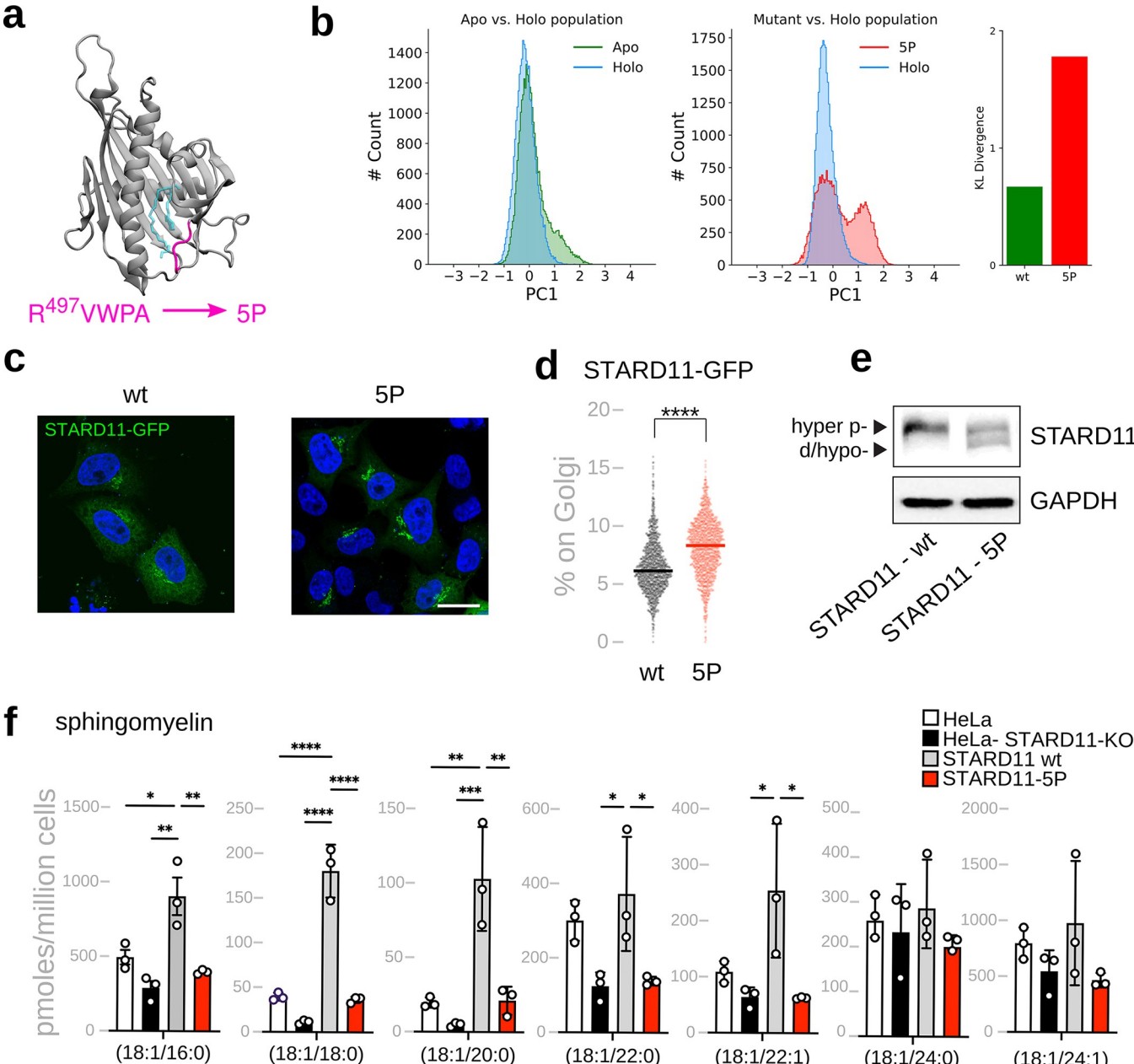

**Fig 6. Mutations altering LTD dynamics impair their lipid transport properties (a)** Design of the STARD11-5P mutant at the omega loop (pink). The position of the lipid in the crystal structure is shown in transparent blue licorice. **(b)** Population distributions of wt apo vs. wt holo states as well as 5P mutant apo vs. wt holo states with their corresponding average KL divergence values. The mutation decreases the similarity between the apo and holo states. **(c)** STARD11-GFP WT (wt) and the STARD11-5P-GFP (5P) mutant localization in HeLa cells analysed by confocal microscopy. Scale bar, 20 μm. **(d)** Percentage of STARD11-GFP WT (wt) and the STARD11-5P-GFP (5P) mutant associated with the Golgi complex in HeLa cells. Cells were stained with Hoechst and anti-GM130 antibody and analysed by automated fluorescence microscopy ($n > 1,000$ cells per condition; ****$p < 0.0001$ [Student $t$ test]. STARD11-GFP WT grey, STARD11-5P-GFP in red). **(e)** Western blot of HeLa cells expressing STARD11-GFP WT (wt) or the STARD11-5P-GFP (5P) mutant. Hyperphosphorylated (hyper-p-) and de/hypophosphorylated (d/hypo-) bands are indicated by arrowheads. **(f)** Mass spectrometry profile of major sphingomyelins in HeLa cells; Hela cells *STARD11*-KO, and Hela cells *STARD11*-KO overexpressing STARD11-GFP WT (wt) or the STARD11-5P-GFP (5P) mutant. ($n = 3$; data are means ± SD; *$p < 0.05$, **$p < 0.01$; ***$p < 0.001$ [ordinary one-way ANOVA]). The data underlying the graphs shown in the figures can be found in https://doi.org/10.5281/zenodo.12728271. KL, Kullback–Leibler; LTD, lipid transport domain; wt, wild type.

interface of 12 LTDs that belong to different families and have diverse secondary structures. We observed a lack of commonality at their membrane binding interface in terms of structural properties such as the amino acid and secondary structure composition, possibly as a consequence of the diversity of the organellar membrane they bind to.

However, we found that across LTDs, the interface and/or residues adjacent to it exhibited high fluctuations in solution and that they displayed the largest collective motion according to PCA. This observation hints that irrespective of the nature of the transported lipid, the high degree of protein conformational plasticity we observed in membrane proximal regions could potentially facilitate the energetically unfavorable extraction of a lipid from a membrane and its transfer to the hydrophobic groove of an LTP. This process has also been proposed as the rate-limiting step for lipid transport [61], and a recent study on CPTP has proposed that lipid transport is enhanced by the disruption of hydrophobic lipid–membrane contacts [62]. Based on our data, it is tempting to speculate that the ability of LTDs to adopt multiple conformations could further contribute to a reduction in lipid–membrane contacts and, hence, lipid transport. However, the slowdown in diffusion at the water–membrane interface makes it computationally challenging to investigate protein conformational transitions when adsorbed on the membrane.

Furthermore, we established that the conformational landscape of LTDs can be modulated by the bound lipid. Our results not only generalize previous observations made by us [63] and others [11,13,14,16–29,46,64] in the context of individual LTPs but also provide additional clues. For example, the observed relationship between membrane-binding and LTD's lipid-modulated conformational plasticity suggests that residues located at the membrane-binding interface could promote lipid entry via not only the correct positioning of the LTD hydrophobic cavity close to the membrane surface but also by undergoing specific conformational changes to promote lipid uptake/release. We foresee that future studies will further investigate the functional consequence of such observation and, most notably, the characterization of the extent to which such conformational changes affect multiple steps of lipid transport, including membrane binding or lipid extraction and release, and whether these are further modulated when different lipids are being transported.

Mutations that alter lipid transport have often been proposed to do so by changing the affinity for the membrane due to a change in the membrane binding interface [11,65,66]. A key hypothesis arising from our findings is that mutations can also abolish or decrease lipid transfer by affecting the propensity of the protein to transition between apo-like and holo-like conformations. Of note, this mechanistic hypothesis is consistent with the one proposed for several Osh/ORP proteins in which experimental deletion of the N-terminal lid results in defective membrane association and/or lipid transport [11,20,22,67].

Despite providing structural and dynamical insights into the mechanism of LTPs, our approach has limitations. First, our atomistic sampling of the conformational dynamics of LTDs is finite, and even though PCA allows to describe the essential dynamics of the LTDs, we cannot completely describe larger conformational changes that might occur on longer time-scales, such as, for example, the opening/closing of the lid occurring in Osh proteins upon membrane binding [11,14,22]. In addition, while the first principal component (PC1) explains a high percentage of the conformational variance, other, more subtle, conformational features might not be captured by PC1 alone and would require contribution from other principal components. Second, our membrane binding assay, based on CG simulations with the MARTINI model, has 2 major limitations: it requires the use of an elastic network to restrain the secondary structure of the protein [68] and it lacks sensitivity towards the lipid composition of membranes. Hence, this method is not well suited to investigate whether the conformational changes induced by the presence of the lipid inside the LTD cavity alter the membrane binding

propensity and to what extent membrane properties such as electrostatics, curvature, and packing defects play a role in the directionality of the transfer cycle. Deciphering the lipid transfer reaction in silico remains a challenging task, as several mechanisms including protein dynamics, protein–membrane interactions, and protein–protein interactions contribute to the overall process. However, our approach to understand LTD conformational plasticity has greatly facilitated the identification of physiologically relevant regions/amino acids in LTPs that underlie their activity (Fig 4).

The behaviour we observe for LTPs is reminiscent of other, previously described, functional protein mechanisms, including enzyme dynamics during catalysis [69], the alternating-access model of membrane transporters [70], or GPCR dynamics [71]. In all these cases, protein dynamics is strongly coupled to ligand binding [72] and protein function, be it for signaling, transport, or enzymatic activity. Unlike these fields, however, the contribution of structural and spectroscopic studies to uncover LTP dynamics remains quite limited, and our simulations provide an important contribution to fill this gap. We hope that our results will motivate researchers to increase efforts to experimentally quantify LTPs conformational plasticity, for example, by structural determination of LTPs in different states (or bound to different lipids) or by single-molecule spectroscopy studies.

Overall, our work demonstrates the importance of conformational plasticity of LTPs in their mechanistic mode of action and provides a framework to understand their functional mechanism more generally, in a protein-aspecific context. The observation that all LTDs share common conformational features raises interesting questions concerning the evolutionary origin of this apparent convergence. We hope our results will contribute to the progress of this still underappreciated aspect of LTP research.

## Methods

### Coarse grain simulations of protein–membrane systems

The atomistic structure of the LTDs studied were obtained from RCSB PDB [73] (Table 1) and were converted to a CG model using the martinize script. An additional elastic network with a force constant of 700 kJ mol$^{-1}$ nm$^{-2}$, and upper elastic bond cutoff of 0.8 nm, was used to restrain the secondary structure of the protein in its apo structure. The CHARMM-GUI Bilayer Builder for Martini [74] was used to build lipid bilayers with lateral dimensions of 20 nm × 20 nm. The bilayers were then equilibrated according to the standard 6-step equilibration protocol provided by CHARMM-GUI. Water molecules and ions were stripped off the system, and the protein of interest was placed away from the membrane, such that the initial minimum distance between any CG bead of the protein and any CG bead of the membrane was at least 3 nm. The orientation of the protein was such that its principal axes were aligned with the *x*, *y*, and *z* directions of the system, with the longer dimension of the protein along the *z* direction. Despite starting from a single orientation, all proteins undergo extensive tumbling before binding to the bilayer, as illustrated by the angle between the 2 principal protein axes and the membrane normal for the 2 proteins that display the highest binding propensity, GM2A and TTPA (Fig I in S1 Text). This setup was then solvated and ionized with 0.12 M of sodium and chloride ions to neutralize the system and reproduce a physiological salt concentration.

Eight independent replicas of 3 μs each were simulated for each LTD–membrane system using the GROMACS [75] (2018.x-2021.x) package and the Martini 3 force field [76]. Energy minimization was performed using the steepest descent algorithm, followed by a short MD run of 250 ps. Production runs were performed at a temperature of 310 K using a velocity-rescale thermostat [77], with separate temperature coupling for protein, bilayer, and solvent

particles and a coupling time constant of 1.0 ps. The md integrator was used for the production runs, with a time step of 20 fs. The Parrinello–Rahman barostat [78] was used to maintain the pressure at 1 bar, with a semi-isotropic pressure coupling scheme and a coupling time constant of 12.0 ps. The Coulombic terms were computed using reaction-field [79] and a cutoff distance of 1.1 nm. A cutoff scheme was used for the VdW terms, with a cutoff distance of 1.1 nm and Verlet cutoff scheme for the potential-shift [80]. The Verlet neighbor search algorithm was used to update the neighbor list every 20 steps with a buffer tolerance of 0.005 $\text{kJ mol}^{-1}\ \text{ps}^{-1}$. Periodic boundary conditions were used in all 3 directions. The system setup and simulation parameters are in accordance with the recently proposed protocol to study transient protein–membrane interactions with the Martini force field [30].

## Atomistic simulations of protein in solution

The CHARMM-GUI Solution Builder [82] was used to construct setups of the protein in solution in the apo (without lipid) and holo (lipid-bound) forms, as indicated in Table 1. Starting structures for mutant apo-form and holo-form simulations (Table 1) were prepared using the CHARMM-GUI Solution Builder [82] starting from the corresponding wild-type apo and holo structures. The stoichiometry of protein to lipids were one-to-one for all LTDs in the dataset except for the SMP domain of E-Syt2. The electron density maps of the SMP domain show clear densities for 2 lipid-like molecules per E-Syt2 monomer [83]. One density is consistent with that of a dioleoylglycerol (DOG) lipid, while the other is reminiscent of Triton X-100, a detergent used in protein purification. Denaturing mass-spectrometry, however, shows that 2 lipid molecules can bind per monomer [83]. Hence, we replaced the 2 detergent molecules in the domain with a DOG each, by aligning a vector from the head-group to tail of the reminiscent detergent with that of a DOG molecule. Thus, in total, 4 molecules of DOG were used for simulations of the E-Syt2 SMP in the holo-form. Standard CHARMM-GUI lipid parameters were used for DOG, oleic acid, cholesterol, ergosterol, DOPC, DOPE, and DOPS lipids, while parameters for ceramide-1-phosphate were obtained from the CGenFF web server [84] and parameters for α-tocopherol were determined by using the structure density factors in the PDB structure when constructing the system using the CHARMM-GUI Solution Builder. A cubic box of edge length was constructed such that the distance between the protein and the edge of the box was at least 1 nm, and the systems were solvated with CHARMM TIP3P water and ionized with 0.12M of sodium and chloride ions. Six independent replicas of 500 ns each were simulated for each system using the GROMACS [75] (2018.x-2021.x) package and the CHARMM36m force field [85]. Energy minimization was performed using the steepest descent algorithm and was followed by a short NVT and NPT equilibration of 100 ps each with position restraints on the backbone atoms of the protein. Production runs were performed using a velocity-rescale thermostat [77] at a temperature of 310 K, with separate temperature coupling for protein and solvent particles, and a coupling time constant of 0.1 ps. The first 10 ns of the production runs of each replica were not considered for analysis. The md integrator was used for the production runs, with a time step of 2 fs. The pressure was maintained at 1 bar by using the Parrinello–Rahman barostat [78] and an isotropic pressure coupling scheme with a compressibility of $4.5 \times 10^{-5}\ \text{bar}^{-1}$ and a coupling time constant of 2.0 ps. The Particle Mesh Ewald (PME) methods was used to compute the electrostatic interactions with a Fourier spacing of 0.16 nm, a cutoff of 1.2 nm, and an interpolation order of 4 was used. Van der Waals (VDW) interactions were switched to 0 over 10 to 12 Å. The LINCS algorithm was used to constrain bonds involving hydrogen atoms. Periodic boundary conditions were employed in all 3 directions.

To confirm the conformational differences between apo and holo states, simulations of the protein in the apo-form were performed for E-Syt2, Sfh1, and TTPA, by starting from the final structure obtained from the holo simulations by removing the lipid from the last frame of the simulations (500 ns × 6 replicas). The sampling and PC obtained in this case are similar to the ones obtained from simulating the apo form of the LTD starting from the crystal structure, indicating that all conformational changes observed are reversible and the protein is not trapped in a metastable state (Fig J in S1 Text).

## Analysis

**Membrane binding interface from CG simulations.** The membrane-interacting residues of each protein were determined by computing the longest duration of interaction of each residue with the bilayer using the Prolint [86] package.

The frequency of interaction with the membrane for each residue as shown in Figs 2B and 3A corresponds to the normalized longest duration of interaction for that residue, obtained from the Prolint package, with a cutoff distance of 7 Å used to define the protein–lipid interactions. The normalized binding frequencies for each protein were obtained by dividing all the longest duration of interaction data by its corresponding largest value. Similarly, the frequency of interaction for each amino acid and each secondary structure type in the LTD (Fig 1D and 1E) were determined by summing the longest duration of interaction for each amino acid/secondary structure type within the protein and obtaining normalized values between 0 and 1 for every protein. The mean interaction across all proteins was then determined by averaging over these normalized values.

## Principal component analysis and clustering from atomistic simulations

To analyze and compare the conformational changes that occur in the simulations of the apo and holo forms of each protein, we first performed PCA of the protein's Cα carbon atoms using the Scikit-learn package [87] (version 0.21.3). The proteins were aligned, and the analysis was performed on the entire and concatenated set of simulations for each protein (6 apo and 6 holo). The highly mobile N- and C-terminal ends of the proteins (2 to 5 residues, where necessary), as well as highly mobile residues 533 to 543 in STARD11, which correspond to residues 507 to 517 in the crystal structure, were excluded from the analysis to avoid possible biasing of the PCs. In the case of STARD11, also the first 58 residues at the N-terminal were discarded, as they correspond to a helix that is normally not considered part of the domain. After scrutiny of the resulting PCs, we focused our attention on the first PC, i.e., PC1 (the PC with the highest variance), which showed most often a consistent convergence with the results obtained when the apo and holo simulations were analyzed separately. The projection of the protein dynamics along the analyzed PCs was further clustered using CLoNe [43], and the relative apo and holo populations of each cluster were calculated by determining how many frames of each cluster were derived from the corresponding simulations. The KL divergence between the apo and holo distributions was computed using the SciPy package [88] (version 1.11.4). The final reported value is the average between both divergence values (apo to holo, holo to apo).

## Root mean square fluctuations

The RMSF analysis of the protein's Cα carbon atoms was done using the built-in GROMACS rmsf tool over the aligned concatenated apo + holo trajectories, taking the same set of residues as for the PCA. The reference structure was taken as the same structure used for the alignment of the trajectories.

## Statistical tests

The RMSF values and the PC1 contributions of membrane-binding and non-membrane-binding residues were compared with a Mann–Whitney U nonparametric test using SciPy [88] (version 1.11.4). The membrane-binding residues were considered as those with a membrane binding frequency greater than 0.3, after normalizing.

## Cavity volumes

Cavity volumes were estimated using the mdpocket tool of the fpocket package [89] selecting only the internal protein cavity for the calculation.

Secondary structure images of the protein were rendered using VMD [90] or ngl viewer [91].

## STARD11 experimental investigations

**Cell lines and culture conditions.** HeLa cells were grown in DMEM high glucose, GlutaMAX (Gibco, USA) supplemented with 10% (v/v) foetal bovine serum (FBS), 4.5 g/L glucose, 2 mM L-glutamine, 1 U/mL penicillin and streptomycin under controlled atmosphere (5% $CO_2$ and 95% air) at 37˚C.

## Plasmid transfection

Human STARD11 WT and mutants were inserted into pEGFP-C1 for WT or pcDNA3.1 eGFP vector for mutants to produce protein with eGFP at the N-terminus. Plasmids were transfected into HeLa cells with jetPRIME transfection reagent (Polyplus Transfection, 114–15) following the manufacturer's instructions.

## Antibodies

The following primary antibodies were used: rabbit anti-COL4A3BP (Sigma Aldrich, HPA035645, RRID: AB_10600700, 1:5,000 for WB, 1:300 for IF), rabbit anti-GOLPH3 (Abcam ab98023, RRID: AB_10860828, 1:300), mouse anti-GAPDH Clone 6C5 (Santa Cruz Biotechnology, sc-32233, RRID: AB_627679, 1:2,000). The following secondary antibodies were used: donkey A568-conjugated anti-rabbit (Thermo Fisher Scientific, A-10042, RRID: AB_2534017, 1:400), donkey A647-conjugated anti-mouse (Thermo Fisher Scientific, A-31571, RRID: AB_162542, 1:400 or Jackson ImmunoResearch, 715.605.150, AB_2340862, 1:200 for quantitative image analysis), donkey HRP-conjugated anti-rabbit (Jackson ImmunoResearch, 711-035-152, RRID: AB_10015282, 1:10,000), and donkey HRP-conjugated anti-mouse (Jackson ImmunoResearch, 715-035-150, RRID: AB_2340770). Hoechst was purchased from Life Technologies (H3570, 1:10,000 from 10mg/mL stock).

## Immunofluorescence, staining, and image analysis

HeLa cells were grown on glass coverslips, treated according to the experimental procedure, fixed with 4% paraformaldehyde for 15 min at RT, and washed 3 times with PBS. After fixation, cells were blocked with 5% BSA and permeabilized with 0.5% saponin for 20 min at RT, followed by 1-h incubation with selected antibodies against the antigen of interest in blocking solution. Cells were then washed 3 times with PBS and incubated with appropriate isotype-matched, Alexa Fluor–conjugated secondary antibodies diluted in blocking solution for 30 min. After immunostaining, cells were washed 3 times in PBS and once in water, to remove salts. After Hoechst staining for nuclei, the samples were mounted with Fluoromount-G (Southern Biotech, 0100–01) on glass microscope slides and analysed under a confocal

microscope Leica SP8 with 63× oil objective (1.4 NA) or Zeiss LSM700 with 40× air objective (1.3 NA). Optical confocal sections were taken at 1 Airy unit under nonsaturated conditions with a resolution of 1,024 × 1,024 pixels and frame average 4. Images were then processed using Fiji software (https://imagej.net/Fiji) [79].

## Quantitative cell imaging

HeLa cells were seeded in a μ-Plate 96 Well Black (IBIDI, 89626) at a concentration of $8 \times 10^4$ cell/well for 48-h transfection experiment or $1.2 \times 10^4$ cells/well for 16-h transfection experiment. STARD11-GFP WT or mutant plasmids were transfected using the TransIT-X2 Transfection Reagent (Mirus Bio, MIR 6000). Cells were fixed in 3% paraformaldehyde for 20 min before staining with antibodies diluted in a PBS solution of 1% BSA and 0.05% saponin. After washing with an automated plate washer (BioTek EL406), cells were incubated for half an hour with appropriate secondary antibodies and nuclei were stained by Hoechst. Cells were left in PBS and imaged by ImageXpress Micro Confocal microscope (Molecular Devices, Sunnyvale, CA); for each well, 49 frames were taken in widefield mode with a 40× objective. Images were quantified using MetaXpress Custom Module editor software to first segment the image and generate relevant masks, which were then applied on the fluorescent images to extract relevant measurements.

## SDS-PAGE and western blotting

After treatment, the cells were washed 3 times with PBS and lysed in a buffer consisting of 20 mM MOPS (pH 7.0), 2 mM EGTA, 5 mM EDTA, 60 mM β-glycerophosphate, 30 mM NaF, 1 mM $Na_3VO_4$, 1% (v/v) Triton X-100, phosphatase inhibitor (PhosSTOP, Sigma-Aldrich), and protease inhibitor (cOmplete Protease Inhibitor Cocktail, EDTA free, Roche). The lysates were clarified by centrifugation and quantified with Pierce BCA Protein Assay Kit (Thermo Fisher Scientific) according to the manufacturer's instructions. Samples were prepared by adding 4× SDS sample buffer, denatured at 95°C for 5 min, and resolved by SDS-PAGE and immunoblot. For immunoblotting, the membrane strips containing the proteins of interest were blocked in TBS-T/5% BSA for 45 min at RT and then incubated with the primary antibody diluted to its working concentration in the blocking buffer for 1 h at RT. After washing with TBS-T, the strips were incubated for 1 h with the appropriate HRP-conjugated secondary antibody diluted in the blocking buffer. After washing with TBS-T, the strips were incubated with the ECL solution for 3 min and exposed to X-ray films, which were then scanned. The intensity of the bands and preparation of images was done using Fiji [92] and Adobe Illustrator 2020.

Sphingolipidomics analysis carried out as described in [93]. Frozen cell pellets were resuspended in 50 μL PBS and extracted with 1 mL Methanol/MTBE (methyl-tert-butyl ether)/Chloroform (MMC) [4:3:3; (v/v/v)] at 37°C (1,400 rpm, 30 min). Internal lipid standards include $D_7$SA (d18:0), $D_7$SO (d18:1), dhCer (d18:0/12:0), ceramide (d18:1/12:0), glucosylceramide (d18:1/8:0), SM (d18:1/18:1($D_9$)), and $D_7$-S1P. The single-phase supernatant was collected, dried under $N_2$, and dissolved in 70 μL methanol. Untargeted lipid analysis was performed on a high-resolution Q Exactive MS analyzer (Thermo Fisher Scientific) as described earlier [93].

## Supporting information

**S1 Text. Supporting information. Table A.** Total variance from the PCA of all the proteins considered in this work along with the contributions from the 2 first principal components. **Fig A.** Time trace of minimum distance values between the protein and the bilayer, for each replica of simulation; indicates transient and reversible interactions. **Fig B.** Residue-wise

frequency of interaction with the lipid bilayer. Residues that have been experimentally proposed to be crucial for membrane binding are highlighted in blue. **Fig C.** Interaction frequency of each amino acid with the lipid bilayer, shown for each LTP in our dataset. **Fig D.** Residue-wise contribution to principal component PC1 for the apo (green), holo (blue), and combined (red) trajectories. GM2A and LCN1 were simulated in the apo-form alone due to the lack of a lipid-bound crystal structure of the protein. **Fig E.** Comparison of the apo-like (orange) and holo-like (purple) structures of the LTDs arising from the extreme ends of the clustering procedure, and the RMSD between them. Osh4 and Osh6 not shown as the N-terminal lid of the protein that exhibits the largest motion can exist in several folded and unfolded states resulting in a wide range of RMSD values. **Fig F.** Cavity volumes for apo (orange) and holo (purple) forms of the protein, computed on the projections of PC1. The cavity volumes for the different clusters (C1, C2, and C3) follow the same colour scheme and notations as in Fig 4C. **Fig G.** Comparison between the overlap of apo-wt (blue)–holo-wt (green) distributions and apo-mutant (red)–holo-mutant (orange) distributions for all mutant proteins considered in this work. The KL divergence values are depicted on the right side. **Fig H.** PCA of mutants. Residue-wise contribution to principal component PC1 for the wt-apo (green), wt-holo (blue), and mutant-apo (red) for STARD11, GRAMD1A, CPTP, and PITPA. **Fig I.** Angle between the membrane normal and the 2 principal protein axes (red and orange) in CG-MD simulations of membrane binding for GM2A (left) and TTPA (right) along with the minimum distance values between the proteins and the bilayer (gray) for all replicas All values have been smoothed by block-averaging every 2 ns. **Fig J.** The population distributions of PC1 from simulations of apo form (orange), holo form (purple), and simulations of the protein in the lipid-less-form starting from the final structure obtained from the holo simulations by removing the lipid from the last frame of the simulations (green). The PC obtained from resampling (green) are similar to the ones obtained from simulating the apo form of the LTD starting from the crystal structure (orange), indicating that all conformational changes observed are reversible and the protein is not trapped in a metastable state.
(DOCX)

**S1 Raw Images. Raw images.**
(PDF)

## Acknowledgments

We especially thank Sebastian Streb of the FGCZ Metabolomics division for excellent technical guidance. The authors thank P. Campomanes for critical reading of the manuscript. We thank Dimitri Moreau and Stefania Vossio (ACCESS Geneva) for microscopy and data analysis.

## Author Contributions

**Conceptualization:** Sriraksha Srinivasan, Andrea Di Luca, Daniel Álvarez, Arun T. John Peter, Giovanni D'Angelo, Stefano Vanni.

**Data curation:** Sriraksha Srinivasan, Andrea Di Luca.

**Funding acquisition:** Stefano Vanni.

**Investigation:** Sriraksha Srinivasan, Andrea Di Luca, Daniel Álvarez, Arun T. John Peter, Charlotte Gehin, Museer A. Lone, Stefano Vanni.

**Methodology:** Sriraksha Srinivasan, Andrea Di Luca, Daniel Álvarez.

**Project administration:** Stefano Vanni.

**Supervision:** Thorsten Hornemann, Giovanni D'Angelo, Stefano Vanni.

**Validation:** Giovanni D'Angelo.

**Visualization:** Sriraksha Srinivasan, Andrea Di Luca, Charlotte Gehin, Giovanni D'Angelo, Stefano Vanni.

**Writing – original draft:** Sriraksha Srinivasan, Arun T. John Peter, Giovanni D'Angelo, Stefano Vanni.

**Writing – review & editing:** Sriraksha Srinivasan, Andrea Di Luca, Daniel Álvarez, Arun T. John Peter, Museer A. Lone, Giovanni D'Angelo, Stefano Vanni.

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
