## [Editor Report · Decision Letter 0]

17 Sep 2023

Dear Dr Vanni, 

Thank you for submitting via Review Commons your manuscript entitled "Conformational dynamics of lipid transfer domains provide a general framework to decode their functional mechanism." for consideration as a Research Article by PLOS Biology.

Your manuscript has now been evaluated by the PLOS Biology editorial staff as well as by an academic editor with relevant expertise and I am writing to let you know that we would like to consider a revision of your submission, according to your plan, for publication at PLOS Biology. However, we would need you to complete the metadata before be invite you to sumit a revision (see below).

To complete the metadata, please login to Editorial Manager where you will find the paper in the 'Submissions Needing Revisions' folder on your homepage. Please click 'Revise Submission' from the Action Links and complete all additional questions in the submission questionnaire.

Once your full submission is complete, your paper will undergo a series of checks and, after your manuscript has passed the checks, we will send you the decision inviting you to submit a revision. To provide the metadata for your submission, please Login to Editorial Manager (https://www.editorialmanager.com/pbiology) within two working days, i.e. by Sep 19 2023 11:59PM.

Kind regards,

Ines

--

Ines Alvarez-Garcia, PhD

Senior Editor

PLOS Biology

---

## [Editor Report · Decision Letter 1]

19 Sep 2023

Dear Dr Vanni,

Thank you for submitting via Review Commons your manuscript entitled "Conformational dynamics of lipid transfer domains provide a general framework to decode their functional mechanism."

After evaluating the manuscript, the reviews and the revision plan, we would like to invite you to revise the work to thoroughly address the reviewers' reports.

**IMPORTANT - SUBMITTING YOUR REVISION**

3. Resubmission Checklist

a) *PLOS Data Policy*

b) *Published Peer Review*

d) *Blurb*

Please also provide a blurb which (if accepted) will be included in our weekly and monthly Electronic Table of Contents, sent out to readers of PLOS Biology, and may be used to promote your article in social media. The blurb should be about 30-40 words long and is subject to editorial changes. It should, without exaggeration, entice people to read your manuscript. It should not be redundant with the title and should not contain acronyms or abbreviations. For examples, view our author guidelines: https://journals.plos.org/plosbiology/s/revising-your-manuscript#loc-blurb

Sincerely,

Ines

--

Ines Alvarez-Garcia, PhD

Senior Editor

PLOS Biology

---

## [Decision Letter · Decision Letter 2]

22 Mar 2024

Dear Dr Vanni,

Thank you for your patience while we considered your revised manuscript entitled "Conformational plasticity of lipid transfer domains characterizes their functional mechanism" for consideration as a Research Article at PLOS Biology. Your revised study has now been evaluated by the PLOS Biology editors, the Academic Editor and the three original reviewers. 

The reviews are attached below. As you will see, the reviewers are now mostly satisfied with the revision, however Reviewer 2 still raises one point that would need to be addressed. This reviewer thinks that you should add comparisons to mutant-holo to support the main conclusion and, along with Reviewer 1, also asks for several clarifications.

In light of the reviews, we are pleased to offer you the opportunity to address the remaining points from the reviewers in a revision that we anticipate should not take you very long. We will then assess your revised manuscript and your response to the reviewers' comments with our Academic Editor aiming to avoid further rounds of peer-review, although might need to consult with the reviewers, depending on the nature of the revisions.

**IMPORTANT - SUBMITTING YOUR REVISION**

3. Resubmission Checklist

a) *PLOS Data Policy*

b) *Published Peer Review*

d) *Blurb*

Please also provide a blurb which (if accepted) will be included in our weekly and monthly Electronic Table of Contents, sent out to readers of PLOS Biology, and may be used to promote your article in social media. The blurb should be about 30-40 words long and is subject to editorial changes. It should, without exaggeration, entice people to read your manuscript. It should not be redundant with the title and should not contain acronyms or abbreviations. For examples, view our author guidelines: https://journals.plos.org/plosbiology/s/revising-your-manuscript#loc-blurb

Sincerely,

Ines

--

Ines Alvarez-Garcia, PhD

Senior Editor

PLOS Biology

Reviewers' comments

Rev. 1:

Having previously reviewed this manuscript positively (Reviewer 1 in the first iteration), my current feedback will be concise and focused.

Srinivasan et al. present a comprehensive study on the structure-dynamics-function relationship of lipid transfer proteins through extensive molecular simulations complemented by some experimental validation.

The research is both timely and significant.

The methodologies and strategies employed are well-chosen and described in sufficient detail for reproducibility.

The results are clearly presented and thoughtfully discussed.

The conclusions drawn are supported by the evidence.

The revisions, mainly prompted by feedback from the review process, have notably enhanced the manuscript, with deeper simulation analyses adding substantial value.

While, in principle, I endorse the acceptance of this article in its present form, I offer two points for consideration:

1. Journal Suitability

Although the manuscript has been strengthened through revisions, it has shifted its focus towards computational analysis, particularly following the removal of experimental data on Mdm12 due to the non-mechanistic signature in its functionality.

This shift renders the study more aligned with computational biology, suggesting PLOS Computational Biology as a potentially more fitting publication venue.

Though I proceed with a "Minor Revision" recommendation (the system does not allow me to offer another suitable recommendation unambiguously), I highlight this consideration regarding journal alignment for the authors' and editors' contemplation.

2. Clarification on Simulation-Experiment Agreement

There appears to be a contradiction in the manuscript regarding the agreement between simulations and experimental findings, particularly concerning the membrane-binding interfaces.

While initial statements highlight an "excellent agreement" (page 5, last paragraph), subsequent discussion regarding Osh6 (page 10) indicates discrepancies potentially attributed to simulation limitations in capturing extensive conformational changes.

I suggest the authors refine their narrative to accurately reflect these nuances and ensure consistency in their discussion of simulation-experiment agreement.

Rev. 2:

The authors have substantially improved the manuscript through their revisions. I have one remaining concern that needs to be addressed before I can support publication and a few minor comments/suggestions:

My main concern regards the investigations of newly added mutants (GRAM1DA-5P, CPTP V158N, PITPA P78L) and STARD11-5P mutant to test the hypothesis that overlap between apo and holo conformational distributions informs on lipid transport rates. To test this hypothesis, the conformational distributions of mutant-apo and mutant-holo should be compared and their overlap found to be smaller than the corresponding overlap between wt-apo and wt-holo distributions. Currently the mutant-apo is compared to the wt-holo distribution; however, this does not rule out the possibility that the mutation alters the conformational distribution of the holo form in such a way to reduce the overlap (i.e., the present analysis assumes the mutation does not impact the holo conformations). The comparisons to mutant-holo are necessary to support the authors' main conclusion.

Fig 2 presents normalized RMSFs. Are the RMSF magnitudes similar across all proteins investigated or are there LTPs that are overall more rigid or flexible than others?

PC1 certainly explains the most variance, albeit typically less than 50%, and well captures differences in conformational changes that seem functionally relevant. Residue's contributions to PC1 are strikingly similar to their RMSFs (Fig 2b vs Fig 3a). Differences in PC1 do not fully explain differences in cavity volume of apo vs holo forms (Fig S6), suggesting that some relevant conformational features may not be captured by PC1 alone. For these reasons, it would be great if the authors could comment about and/or illustrate the features captured by PC2 in the SI.

I appreciate the authors edits to clarify their focus on conformational plasticity, a thermodynamic property. Nevertheless, on p. 8, PCA is described as filtering out "high-frequency fluctuations occurring during the protein dynamics" to "identify… slower motions," and "explaining most of the variance in the dynamics." While PCA is evocative of this (and often described with similar phrasing in the literature), it does not explicitly account for temporal correlations and dynamics, such as done in tICA. To avoid misleading readers (especially given the increasingly imprecise and ambiguous use of the terms "conformational ensemble" and "dynamics" in the current biomolecular modeling literature), I suggest the authors revise these statements to emphasize PCA's utility for capturing dominant, large-scale variations in protein conformations sampled during the simulations.

It would aid readers if the authors specifically state in the abstract that the "common feature" is that LTPs interconvert between different conformational states (or a similar phrasing equating this feature to their conformational plasticity).

Please increase the size of all axes' labels and tick marks in Fig 2c, 3b, bar plots of KL divergence shown in Fig 5 and 6b, and Fig S1.

On p. 3, I suggest the wording "transferring a lipid between two membranes" would be preferable to "across two membranes" to avoid any confusion with lipid flip-flop.

On p. 10, I suggest the wording "regions… are involved in large collective motions" over "display" to emphasize membrane binding regions' contributions to conformational changes extending over the entire protein, not just membrane-bound regions, as highlighted by the authors in their response.

A few minor grammatical issues need addressing in (i) the abstract: "allowing to interpret and design experimental strategies"; (ii) p. 3: "allowed to fine-tune the mechanism"; (iii) p. 4: "implications with" should be "implications for"; and (iv) p. 13: "on LTD conformational landscape" should be "on a LTD's…".

Rev. 3: Antoine Taly - note that his reviewer has signed his review

The authors have done a good job in using the review to improve the paper!

---

## [Editor Report · Decision Letter 3]

15 May 2024

Dear Dr Vanni,

Thank you for your patience while we considered your revised manuscript entitled "Conformational plasticity of lipid transfer domains characterizes their functional mechanism" for publication as a Research Article at PLOS Biology. This revised version of your manuscript has been evaluated by the PLOS Biology editors and the Academic Editor.

Based on our Academic Editor's assessment of your revision, we are likely to accept this manuscript for publication, provided you satisfactorily address the data and other policy-related requests stated below.

In addition, we would like you to consider a suggestion to improve the title:

"The mode of action of structurally unrelated lipid transport proteins correlates with their conformational plasticity"

We expect to receive your revised manuscript within two weeks. 

*Published Peer Review History*

*Press*

Sincerely,

Ines

--

Ines Alvarez-Garcia, PhD

Senior Editor

PLOS Biology

Fig. 1D, E; Fig. 2B, C; Fig. 3A, B; Fig. 4B, C; Fig. 5A, B; Fig. 6B, D, F; Fig. S1; Fig. S2; Fig. S3; Fig. S4; Fig. S6; Fig. S7; Fig. S8; Fig. S9 and Fig. S10

***While we can see that you have deposited some of the data underlying the graphs in Zenodo (https://doi.org/10.5281/zenodo.7819506), we would like you to clarify the relationship between the zenodo files and the figures, labelling the data to the corresponding graphs and noting this in each figure legend. Please also add in an excel file any missing data that it is not deposited in Zenodo.

We require the original, uncropped and minimally adjusted images supporting all blot and gel results reported in an article's figures or Supporting Information files - mainly in Fig. 6E. We will require these files before a manuscript can be accepted so please prepare and upload them now. Please carefully read our guidelines for how to prepare and upload this data: https://journals.plos.org/plosbiology/s/figures#loc-blot-and-gel-reporting-requirements

CODE POLICY

Per journal policy, if you have generated any custom code during the curse of this investigation, please make it available without restrictions upon publication. Please ensure that the code is sufficiently well documented and reusable, and that your Data Statement in the Editorial Manager submission system accurately describes where your code can be found. [IF APPLICABLE: As the code that you have generated to XXX is important to support the conclusions of your manuscript, its deposition is required for acceptance.]

---

## [Editor Report · Decision Letter 4]

5 Jul 2024

Dear Dr Vanni,

Thank you for the submission of your revised Research Article entitled "The conformational plasticity of structurally unrelated lipid transport proteins correlates with their mode of action" for publication in PLOS Biology. On behalf of my colleagues and the Academic Editor, Rebecca Haeusler, I am delighted to let you know that we can in principle accept your manuscript for publication, provided you address any remaining formatting and reporting issues. These will be detailed in an email you should receive within 2-3 business days from our colleagues in the journal operations team; no action is required from you until then. Please note that we will not be able to formally accept your manuscript and schedule it for publication until you have completed any requested changes.

PRESS

Sincerely, 

Ines

--

Ines Alvarez-Garcia, PhD

Senior Editor

PLOS Biology
